# "Merrily to Hell Together": Threats of Self-Destruction among Golden Age Pirates

## Matthew J. McLaine

Independent Researcher, Ladson, SC 29456, USA; mattmclaine@gmail.com

**Abstract:** The threat of death hung over every aspect of pirate life during the Golden Age of piracy. They threatened governors and governments who dared to capture, prosecute, and hang their fellow buccaneers. They threatened their victims for running away, for fighting back, or for hiding their money. They even threatened death on each other should any of them suggest leaving off their chosen course or for betraying their company. Even the iconic skull and crossbones "Jolly Roger" pirate flag was a visible, physical symbol of a threat of death: for victims it was a reminder that surrender may mean mercy, but resistance would be fatal; and for the Pirates themselves, a grim reminder that capture or failure could mean their end. Many pirate crews in the Golden Age took this menace of death to the extreme by threatening to blow up their ship to avoid the noose, promising to take prisoners and pirates, captives and captors, and gold and galleon to the bottom of the ocean, going "merrily to Hell together". Yet despite their boasts and despite embracing the symbols of death, when the time came to make good on their oaths, few of these crews took that final explosive step and fewer still succeeded. This paper examines twenty incidents from the Golden Age of piracy in which pirates or their victims threatened or attempted to blow up their ships and themselves to avoid capture. Witness statements, period newspaper accounts, and trial testimony reveal that the threat was frequent but the attempt was not. In the end it was often prevented by the pirates themselves after a change of heart, despite promising one another that they would "live & dye together".

**Keywords:** pirates; golden age of piracy; suicide; Jolly Roger; Blackbeard



## 1. Introduction

The threat of death hung over every aspect of pirate life during the Golden Age of piracy. They threatened governors and governments who dared to capture, prosecute, and hang their fellow buccaneers. They threatened their victims for running away, for fighting back, or for hiding their money. They even threatened death on each other should any of them suggest leaving off their chosen course or for betraying their company. Even the iconic skull and crossbones "Jolly Roger" pirate flag was a visible, physical symbol of a threat of death: for victims it was a reminder that surrender might mean mercy, but resistance would be fatal; and for the pirates themselves, a grim reminder that capture or failure could mean their end. Many pirate crews in the Golden Age took this menace of death to the extreme by threatening to blow up their ship to avoid the noose, promising to take prisoners and pirates, captives and captors, gold and galleon to the bottom of the ocean, going "merrily to Hell together" (Leeson 2009, p. 118). This paper opens with a brief look at the attitudes toward suicide in the Golden Age period[1] as well as a review of the role of threats in pirate life, followed by an examination of twenty incidents from the Golden Age of piracy in which pirates or their victims threatened or attempted to blow up their ships and themselves to avoid capture. Witness statements, period newspaper accounts, and trial testimony reveal that the threat was frequent but the attempt was not, and success even less so. In the end it was often prevented by the pirates themselves after a change of heart, despite promising one another that they would "live & dye together".

## 2. Pirates and Suicide

Threatening self-destruction meant not merely accepting death, but death by suicide. This was no stretch, though: pirates were intimately familiar with suicide. George Lowther was one such. An associate of Spriggs, he did what Spriggs had only bragged of: "The Master of the *Eagle* was afterwards informed, that George Lowther, the Captain of the said Pyrate Sloop, had shot himself on the said Island of Blanco, and was found dead with his Pistol burst by his side".[2] Another was found among the crew of John Phillips, whose vessel was taken from him when the captives aboard rose up and retook their ship from Phillips and his pirate crew in 1724. Nearing Boston, they fired a signal gun to alert the harbor fort to send their boats for the captors-turned-prisoners: "At this time some of the pirates were on deck, and one of them asked leave to fire another gun, which being granted, he would not swab the gun out nor have the vent stopped, but put in the cartridge, and stood direct before the muzzle to ram it down, by which means the cartridge took fire, and blew him into pieces; it is supposed he did this purposely in order to escape the punishment which he knew must be his lot in case he was carried into the harbor".[3] Phillips' man and Lowther each used suicide as a way to avoid capture and punishment by colonial authorities.

Francis Spriggs, facing capture, swore his intention to end his life rather than become a prisoner: "when he saw the Danger they were in of being taken, upon the Man of War's out sailing them, was afraid of falling into the hands of Justice; to prevent which, he, and one of his Chief Companions, took their pistols, and laid them down by them, and solemnly Swore to each other, and pledg'd the Oath in a Bumper of Liquor, that if they saw there was at last no possibility of Escaping, but that they should be taken, they would set Foot to Foot, and Shoot one another, to Escape Justice and the Halter" (Barnard 1725, p. 13). Whether death came by their own hand, or another's, did not always matter. William Snelgrave spent time as a captive of Howell Davis and Thomas Cocklyn in 1719. He was asked by a pirate, "Whether I was afraid of going to the Devil by a great Shot? For as to his part, he hoped he should be sent to Hell one of these days by a Cannon Ball" Snelgrave answered, "I hoped that would not be my Road" (Snelgrave 1734, p. 210). Marcus Rediker called this attitude "the death wish of pirates" and noted them "cheering, lustily and repeatedly, for their own mutual destruction" (Rediker 2004, p. 151). Cocklyn's man and Spriggs may have been willing or simply boasting but it was left to pirates such as Lowther to take that final step.

Among pirates, suicide was not only assumed to be a (theoretically) proper way to avoid capture but in certain cases may have been prescribed. Pirate "articles"—the written codes of conduct which governed many ships—often contained a clause specifying that pirates who cheated their shipmates or deserted were to be marooned and given a pistol to end their lives. John Phillips' articles were fairly typical and included the rule, "If any Man shall offer to run away, or keep any Secret from the Company, he shall be marroon'd, with one Bottle of Powder, one Bottle of Water, one small Arm, and Shot".[4] In addition to giving the offender a little water, some articles were careful to sentence the guilty to be set "on Shore, not in an uninhabited Place, but Somewhere, where he was sure to encounter Hardships".[5] By leaving the condemned man at least the potential of survival, the pistol was not automatically a tool of suicide, but the possibility remained, and leaving them behind could amount to "a cruel refinement intended to prolong his agony".[6] Their fellow pirates understood the sacrifice inherent in suicide—in the name of avoiding criminal consequences, or as a result of them—but society and the law were not so understanding, either of the pirates' lifestyle or of their choice to end it.

The law was not favorable to suicides or to pirates, much less to suicidal pirates. "Self-murder, the pretended heroism, but real cowardice" wrote Sir William Blackstone of suicide (Blackstone 1775, vol. 4, p. 189). It was an offense against both God in the presumptuous taking of a life, and against the King for assaulting one of his subjects, and so Blackstone noted that "the law has therefore ranked this among the highest crimes, making it a peculiar species of felony, a felony committed on one's self" (Blackstone 1775, vol. 4, p. 189). Because of this double guilt laid upon suicides, English law required that the

guilty party's property be forfeited to the Crown. Commoners—jurors included—were not so high-minded and would sometimes refuse to pass judgment on pirates[7] or on suicides, "routinely declaring all suicides *non compos*" (Bartel 1960, p. 151), and therefore leaving their families immune to asset seizure. Despite jury nullification in suicide cases rising to 97% by 1750, law scholars continued to drive home the "religious wrongfulness" inherent in suicide (Chang 2018, p. 164). Several of the twenty incidents under review here are known to us only from newspaper articles; publicizing these suicidal threats may have been an attempt to portray suicidal pirates as cowards and fools in order to weaken their fearsome reputation (Kazerooni 2016).

Even beyond the religious and social vilification, pirates were already condemned in the law's eyes, and this may have contributed to their willingness to end everything. Piracy was "an offence against the universal law of society; a pirate being according to sir Edward Coke, *hostis humani generis*", wrote Blackstone; "by declaring war against all mankind, all mankind must declare war against him".[8] As such there was limited legal motivation to treat them fairly: at John Quelch's trial the Queen's Admiralty Advocate argued that as "*hostis humani generis*", "neither faith nor oath is to be kept with them" (Rubin 1988, vol. 63, p. 102). Rediker calculated that at least one Golden Age pirate in ten was hanged, and one in four met some sort of violent and premature end: "his was most decidedly not a romantic occupation" (Rediker 2004, p. 163). With mortality and condemnation on their minds, pirates lost all trust in pardons and the promises of authority. One of Bartholomew Roberts' captives reported that the pirates "would have no dealings with Acts of Grace, trepanned under lying promises". He added prophetically, "If they were attacked by too strong a force they would blow up their ships and all go merrily to hell" (Grey 1933, p. 207).

## 3. Life under Threat

That threat may seem extreme, but such threats were part and parcel of pirate life. Peter Leeson's economic study of pirates revealed the extent of threat as a tactic to maximize profit and minimize risk: threatening a merchantman "so intimidates our Sailors that they refuse to fight when the Pirates attack them".[9] The most visible and potent symbol of pirates' threats was the iconic Jolly Roger flag. Its imagery of hourglasses, skulls and skeletons, swords and spears, blood and guns communicated to all viewers (pirates and victims alike) that death awaited any who resisted, time ran short, and the end was near. Ironically, this savage imagery may have prevented further loss of life among their victims by encouraging them to surrender without a fight, furthering at the same time the pirates' economic aims (Leeson 2009, pp. 93, 101). Snelgrave confirmed that such flags were flown specifically "to terrify Merchant-Men", (Snelgrave 1734, p. 4) and pirates were well aware that merely having a Jolly Roger multiplied the threat it represented: such a sight, "they merrily said, would be as good as 50 Men more" (Johnson 1724, p. 417).

Roberts' time on the African coast illustrates how serious both pirates and victims were about their threats. The Governors of Martinique and Barbados sent forces to hunt Roberts before he left the Caribbean and he incorporated into his iconography an implicit threat against them. Approaching the slaving port of Ouidah on the coast of modern-day Benin (formerly the Kingdom of Whydah), they flew twin standards: "the Flag had a Death in it, with an Hour-Glass in one Hand, and cross Bones in the other, a Dart by it, and underneath a Heart dropping three Drops of Blood". (Johnson 1724, p. 260). Simultaneously they hoisted a Jack with "a Man pourtray'd in it, with a flaming Sword in his Hand, and standing on two Skulls, subscribed *A B H* and *A M H*", explained as "signifying a *Barbadian's* and a *Martinican's* Head".[10] Again, it was no idle boast: Roberts' ship "had been a French man of war some small time before taken by Roberts in her way from Martinique to France with the Governor of Martinique on board who the pirates hanged at the yard arm" (Headlam 1933, p. 463). His men would shortly after attempt to make good on their threat to "go merrily to hell" as well, proving that pirate boasts—at least this one's crew—could be taken in deadly earnest.

Henry Morgan threatened to use nuns and priests as human shields when assaulting Porto Bello, "persuaded the governor would not employ his utmost force", but was forced to make good on the threat when the Governor refused to surrender (Exquemelin 1852, p. 105). At Coquimbo, the Governor offered a truce and a glass of wine to Bartholomew Sharp to discuss a ransom for the town; Sharp made a counter offer, threatening to burn the city if the ransom failed to materialize. The town reneged and Sharp's buccaneers carried out their threat, "firing it in several Places at once, and Packing up our Luggage, after we had staid till the greatest part of it was in Flames" (Ayres 1684, pp. 42–44). Blackbeard promised to "burn all Vessels belonging to New-England for executing the Pirates at Boston",[11] and did so in early 1718; the prisoners he took were allowed to leave on another captured vessel, which luckily "belonged to Rhode-Island", and Edward Low fulfilled a threat to destroy shipping out of Boston after the *HMS Greyhound* captured Low's associate Charles Harris. Low and his men brutally tortured and murdered a young master out of Nantucket, "and telling him that because he had been a good captain he should have an easy death, at last they shot him through the head and sunk the sloop" (Dow and Edmonds 1923, p. 209). Pirates may have made threats for economic or reputational reasons, but when pressed they were often willing and capable of carrying out those threats.

## 4. The Twenty Incidents

Of these twenty cases under review, eight never progressed beyond the threat stage. Charles Vane captured the *Neptune* and *Emperor* off the Carolina coast in 1718 and took them to Abaco in the Bahamas to loot both. He was met there by Nicholas Woodall, a smuggler who brought Vane supplies and ammunition as well as information on the defenses at New Providence. Vane took several more vessels in the region before sailing away alongside Woodall. Sailors from the captured ships reported that Vane bragged "that if there came two Men-of-War to attack him, he would fight em and if he could not escape them, he would go into his Powder Room and blow up his ship, and send them on board and himself to Hell together" (Woodard 2007). Vane was later deposed for cowardice and replaced by John "Calico Jack" Rackham; after sailing alongside Robert Deal for a time, Vane was wrecked, captured, tried, and hanged, never having made good on his threat. Deal and Woodall were also captured but neither made the same self-destructive boast as Vane (Johnson 1724, p. 149).

Richard Worley was killed in 1719 off South Carolina when Governor Robert Johnson personally led a flotilla against him. Surprised by a vicious defense from Johnson's fleet (which had disguised the true number of guns they mounted), Worley and his men fought for a time but were overcome. The *Whitehall Evening Post* reported that "this is the very Pirate who boasted he would fight any Man of War, and if he was overcome, would blow up the Conquerors, which being known, Care was taken to prevent it in the boarding his Ship" (*Whitehall Evening Post* 1719). Worley's consort captain John Cole was also captured; neither he nor Worley tried to detonate their ships, perhaps in Cole's case because among his cargo were "some Women, who, during so short a Space of Time in between there being taken by, and retaken from the Pirates, had got Husbands among them" (*Whitehall Evening Post* 1719).

We saw earlier that Francis Spriggs and his fellow pirates were ready to "set Foot to Foot, and Shoot one another, to Escape Justice and the Halter" (Barnard 1725, p. 13). His suicide pact extended to the entire ship as well: Captain Richard Hawkins, captured by Spriggs and crew in 1724, reported that "They have no Thoughts of ever being taken; but swear, with the most direful Imprecations, that if ever they should find themselves over-power'd, they would immediately blow their Ship up, rather than do Jolly Roger the disgrace to be struck, or suffer themselves to be hang'd like Dogs".[12] His crew was eventually captured; various sources claim Spriggs himself escaped justice in the end, or alternately that he was killed by Miskito tribesmen (*Daily Post* 1726). Neither Spriggs nor his pirate co-captain Richard Shipton ended their own lives; their associate Joseph Cooper was another matter which will be covered shortly.

Buccaneer Laurens De Graaf was among five pirate captains who were never forced to destroy their own ships but who, in the thick of hostile action, did threaten or prepare to do so. De Graaf's major exploits pre-date the Golden Age of Piracy by only a few years, though his later raids did extend into the Golden Age period. On a 1685 expedition from Petit-Goâve against the Spanish, he was approached by a small vessel at the head of several others. The newcomer "gave us two guns to make us strike; insomuch that taking him really for a Spaniard, we knocked out the heads of two barrels of powder, in order to burn our selves, and blow up the ship, rather than fall into the hands of those people, who never gave us quarter, but were wont to make us suffer all imaginable torments, they beginning usually with the captain, whom they hang with his commission about his neck".[13] Luckily it was a fellow French buccaneer so their desperate plan never came to fruition. This was in fact De Graaf's standard practice: the previous year he encountered two Spanish frigates near Cartagena and had his powder barrels prepared, again not forced to use them after he turned the tables and overcame both, which earned him the nickname "Lorencillo" among his foes (Thornbury 1858, p. 296).

Another French *corsaire*, Jean Bart was more privateer than pirate and his fame was such that French warships are named in his honor to this day. A 1691 incident during the Nine Years' War saw him board an English vessel and threaten to destroy himself and the ship using the English ship's own gunpowder. The English were paralyzed with fear and Bart's *corsaires* prevailed (Statham 1910, p. 206). The incident was possibly apocryphal but was nevertheless memorialized on trading cards, prints, and etchings into the twentieth century.

Notorious pirate Edward Low was partnering with Charles Harris in 1723 when their ships were surprised by the *HMS Greyhound* off Rhode Island. Low barely escaped, leaving Harris to be captured. Harris' men were individually ready to end their lives to avoid capture, but Low's plan included his ship as well. William Marsh was one of their captives and testified that while he was on Low's ship he observed gunner Thomas Powell, who "seemed to be a brisk, stirring, active Man amongst them, and told the Deponent they always kept a Barrel of Powder ready to blow up the Sloop rather than be taken".[14] While his willingness to torment and execute others multiplied after he learned that Harris had been executed, Low himself never carried through on his threats to go down with his ship (Watson 1830, p. 468).

Though a minor pirate who operated mainly in European waters, John Gow was no stranger to suicidal threats. In early 1725 he tried to negotiate with an old schoolmate named Fea for supplies to repair his ship which ran aground in the Orkney Islands. In a letter Gow threatened "that if the Men of War arrived (for Mr. Fea had given him Notice, that he expected two Men of War) before he was thus assisted, they would set fire to the Ship and blow themselves up; so that as they had lived, they would die together".[15] Gow and his men were captured and hanged before they could enact their threat.

Lynnhaven Bay, Virginia was raided by the pirate Louis Guittar in early 1700. The *HMS Shoreham* under Captain William Passenger caught and defeated Guittar's ship *La Paix*, but was prevented from taking the pirates prisoner because they "had laid a train to thirty barrels of powder and threatened to blow the ship up" (Headlam 1910, p. 523). Guittar's prisoners offered to swim to the *Shoreham* to intercede and Virginia Governor Francis Nicholson—who had embarked on the *Shoreham* for the battle—arranged to refer them to the King's mercy if they would call off their mass suicide and let the prisoners live. At trial, a passenger and several witnesses from among Guittar's prisoners were asked if they heard the pirates threatening to blow up the ship. One witness confirmed, "Yes they did, and they went to prayers upon it". Another swore that "I heard them say they would live and dye together". (*The Trialls of John Hougling, Cornelius Franc, and Francois Delaunee for Piracies and Robberies by Them Committed in a Ship Called the Peace in Company and with the Assistance of Severall Others near the Capes of Virginia* 1700).

Two of our suicidal pirates did attempt to destroy their vessel but were prevented by the prisoners kept aboard. The legendary Edward "Blackbeard" Teach laid careful plans

to avoid capture. When he was finally cornered by Lieutenant Maynard, "Teach had little or no Hopes of escaping, and therefore had posted a resolute Fellow, a Negroe, whom he had bred up, with a lighted Match, in the Powder-Room, with Commands to blow up, when he should give him Orders, which was as soon as the Lieutenant and his Men could have entered, that so he might have destroy'd his Conquerors: and when the Negro found how it went with Black-beard, he could hardly be perswaded from the rash Action, by two Prisoners that were then in the Hold of the Sloop".[16] This African sailor is commonly identified as Caesar, and is often linked with myths of the fictional pirate Black Caesar, though there is no evidence to support the latter (Leigh 2012).

The deposition of Robert Dunn, whose sloop was captured by Bartholomew Roberts in September 1720, brought up the willingness toward self-destruction which would become a hallmark of Roberts' crews: "He was seized by a pirate ship and sloop, commanded by one Roberts, of Barbados, about 130 men all told. . . . They said they intended to take Marygalante [Marie-Galante]. They intend to take their revenge off Antego [Antigua] and Barbados and then go on the coast of Brazil or the East Indies. They would blow up rather than be taken. Every man double armed, and mostly Englishmen" (Headlam 1933, p. 251). Another captive testified, "Whilst I was in the hands of the Pirates nothing was heard from these rascals the whole time but swearing, damning and blaspheming to the last degree imaginable saying they would have no dealings with Acts of Grace, by which to be sent to hang a-sundrying at Hope Point as were the companies of Kidd and Bradish, trepanned under lying promises. If they were attacked by too strong a force they would blow up their ships and all go merrily to hell".[17] Yet another of Roberts' captives swore at trial that Roberts and his close friend George Wilson (who "was very intimate with Roberts") often said "that if they should meet with any of the Turnip Man's Ships, meaning the King's Ships, they would blow up, and go to Hell together" (*A Full and Exact Account, of the Tryal of All the Pyrates, Lately Taken by Captain Ogle, on Board the Swallow Man of War, on the Coast of Guinea* 1723, p. 42).

This was no mere bluster: Roberts' pirates took this sentiment seriously. When Roberts was caught and killed off the African coast in 1722 by Chaloner Ogle aboard the *HMS Swallow*, one of his pirates (James Phillips) ran below decks to the *Fortune's* hold and tried to drop a lighted match into the gunpowder stores. He was seized and stopped by two prisoners, Stephen Thomas and Henry Glasby. Thomas testified, "Stephen Thomas swears, that after the Action with the *Swallow*, and that the *Fortune* had struck her Pyratical Colours, and her Mast down, this Prisoner was down with a lighted Match to blow the Ship up, swearing very prophanely, let's all go to H-ll together, and threw the Deponent against the Ladder, wounding his Hand, as they were struggling about the Match, till Glasby came to his Assistance". He added that Phillips "was ever moross [morose] and drunk, carrying his Pistols sometimes about him, and threaten'd new Comers, if they offered to speak" (*A Full and Exact Account, of the Tryal of All the Pyrates, Lately Taken by Captain Ogle, on Board the Swallow Man of War, on the Coast of Guinea* 1723, p. 70).

It wasn't always up to the prisoners to prevent their own destruction: not all pirates were on board with their comrades' self-destructive urges either. Blackbeard's sometime protégé, sometime prisoner Stede Bonnet was on the brink of defeat and capture by William Rhett when he ordered his gunner George Ross to blow up their ship the *Royal James*. The rest of Bonnet's crew refused, blocked Ross' access to the powder stores, and surrendered. One of the witnesses against Bonnet's men at trial declared "that George Ross, the Gunner of the Pirate's Sloop, was for blowing up the said Sloop, and that he acknowledg'd he was to have set fire to the Train, and that he would have done it" (*Tryals of Major Stede Bonnet and Other Pirates* 1719, p. 18).

That the suicidal sentiment was ingrained in John Gow's crew is clear from a second incident on his ship: in 1724, Gow's first mate Williams accused Gow of cowardice for refusing to attack a heavily-armed vessel. A fight broke out; other pirates shot Williams and "the Men about him laid hold of him to throw him over Board, believing he was dead, but as they lifted him up, he started violently out of their Hands, and leaped directly into

the Hold, and from thence run desperately into the Powder-Room, with his Pistol cock'd in his Hand, swearing he would blow them all up; and had certainly done it, if they had not seiz'd him just as he had gotten the Scuttle open, and was that Moment going in to put his hellish Resolution in Practice".[18]

Edward Low's crew may have boasted of their willingness to go down with their ship, but the crew of Low's consort Charles Harris nearly made it happen. Peter Solgard's *HMS Greyhound* blasted Harris's *Ranger* into submission in a 1723 battle, at the end of which "One desperado was for blowing up this sloop rather than surrendering, and being hindered, he went forward, and with his pistol shot out his own brains" according to a witness quoted in the *Boston News-Letter* (Dow and Edmonds 1923, p. 293). This was not a decision made in panic or haste: as mentioned previously, captive William Marsh noted that Low and Harris' men "always kept a Barrel of Powder ready to blow up the Sloop rather than be taken" (*Tryals of Thirty-Six Persons for Piracy* 1723, p. 9).

Technically a privateering voyage, George Shelvocke's time as Captain of the *Speedwell* was exceptionally well-documented thanks to his own book as well as a rival volume by his Master of Marines, William Betagh. Shelvocke nearly had to put down a mutiny when the temptation to turn from privateering to piracy reared its head, but luckily his crew was not swayed. From Shelvocke's *Voyage Round the World*: "Some time before our arrival here, Turner Stevens, my gunner, very gravely made a proposal to me . . . to go a cruizing in the Red Sea; for, said he, there can be no harm in robbing those Mahometans . . . Upon the hearing of this discourse, I ordered him under confinement; and the man, after that, having threatened in a very outrageous manner, to blow up the ship, I, for these reasons, and others as sufficient, discharged him here, at his request, which I was very glad to hear him make, and to see every body else as well pleased at his departure".[19]

The fire aboard Howell Davis' ship in 1719 was an accidental one, but as we saw previously, some of Davis' and Cocklyn's sailors were men of suicidal impulses. Prisoner William Snelgrave was dining (under threat) with Davis and other pirate leaders off the African coast when a fire broke out in the cargo hold and spread toward the powder stores. The pirate officers were dumbfounded, the sober pirates and prisoners fled, and the rest were too drunk to help. Worse, some of the pirates actively wanted the ship to blow up, fulfilling their suicidal urges: "Whilst I stood musing with my self on the Quarter-deck", Snelgrave recalled in his memoir, "I heard a loud shout upon the Main-deck, with a Huzza, 'For a brave blast to go to Hell with,' which was repeated several times. This not only much surprized me, but also many of the new entered Pirates; who were struck with a Pannick Fright, believing the Ship was just blowing up" (Snelgrave 1734, pp. 266–74). Snelgrave brought back the sober pirates, organized a bucket brigade, and with the help of Richard Taylor ("as brisk and couragious a Man as ever I saw") and other officers, finally saved the ship. He declined when the grateful pirates offered him hearty thanks and plentiful rewards afterwards (Paine 1921, p. 251).

Only two pirate crews actually carried out their self-destructive threats. In February 1722, Bartholomew Roberts sent his lieutenant James Skyrme to retrieve a passing merchant ship with his own *Great Ranger*, but Skyrme was badly wounded and captured when the "merchant" revealed itself as the *HMS Swallow* under Captain Chaloner Ogle. After the dismasted *Ranger* struck its colors and surrendered, six pirates fled below decks where one of them, John Morris, fired a pistol into their remaining gunpowder. Unluckily for the pirates, the *Ranger* had expended most of its powder in the battle; "it was too small a Quantity to effect any Thing more, than burning them in a frightful Manner" (Johnson 1724, pp. 267–68). Morris and one other were killed, and the others were badly burned but the *Ranger* remained intact and was taken by the *Swallow's* marines. The survivors of the attempt were treated but hanged anyway, including Thomas Withstandyenot who confessed "he had been eight Months among these Pyrates . . . and in the Action was wounded by the Powder that blew up in the Steerage; which, as he says, was set on fire by a Pistol by one Morrice, since dead" (*A Full and Exact Account, of the Tryal of All the Pyrates,*

*Lately Taken by Captain Ogle, on Board the Swallow Man of War, on the Coast of Guinea* 1723,
p. 69).

Just one pirate was both prepared and determined enough to succeed in his final
threat. Joseph Cooper's *Night Rambler* was caught in early 1726 by the English frigate
*Diamond*. In a letter from the Bay of Honduras, the *Diamond* reported that they had learned
"of a large Pirate Sloop of eight Guns, and twenty-seven Men, one Cooper Master". The
*Diamond* pursued the pirate and "on boarding her they blew up their Cabin, which did
us the most Mischief, killing one Man, and wounding twelve; and we killed them four,
and wounded twelve" (*London Daily Post* 1726). Cooper was killed; his ship was fitted out
as a pirate hunter and the *Diamond* used it to capture the pirate Richard Shipton and the
remains of Francis Spriggs' crew.

Victims too were capable of choosing self-destruction over self-preservation. Edmund
Congdon (aka Edward "Christopher" Condent) was first mate of a sloop sailing out of
New Providence in 1718 when trouble broke out aboard: "they had among them an Indian
Man, whom some of them had beat; in revenge, he got most of the Arms forward into the
Hold, and designed to blow up the Sloop".[20] Congdon leapt into action, taking a bullet
from the suicidal Indian as he dove in to save the ship, the crew, and himself. Impressed,
the crew chose Congdon for their Captain as they embarked on a spree of piracy spanning
the Caribbean, the African coast, and the Indian Ocean.

Congdon's case involved one disgruntled pirate victimized by his fellow rogues; in
our last two incidents the pirates' victims turned the tables on their attackers. An exciting
story of desperate defense came from the Mediterranean in 1720 when the *Mary and Martha*
was attacked near Malaga off the southern Spanish coast. "[B]eing becalmed in her passage
she was attacked by 3 Sallee Rovers. Upon which the captain ordered some barrels of
gunpowder to be placed in his cabin. and as soon as the Moors to the no. of 50 boarded him
he sprung his mine and blew up above half of them: the crew at the same time laid about
them most manfully with their cutlasses so that the rest were cut to pieces. the Captain
killing 4 with his own hands".[21] The Salé Rovers were successors to the legendary Barbary
coast raiders led by Murat Reis, himself the former Dutch rover Jan Janszoon.

Etienne de Montauban was cruising off the African coast in 1695, alternately flying
Dutch and French colors to lure in potential prey. When he attacked an English guard ship
near Angola his intended victim refused to be taken: "the enemy's powder suddenly taking
fire, by the means of a match the captain had left burning on purpose, as hoping he might
escape with his two shallops, blew both the ships into the air, and made the most horrible
crack that was ever heard".[22] Montauban's account of the explosion is evocative: "there
was formed, as it were, a mountain of water, fire, wreck of the ships, cordages, cannon,
men, with a most terrible clap made, what with the cannon that went off in the air, and the
waves of the sea that were tossed up thither; to which we may add the cracking of masts
and boards, the rending of the sails and ropes, the cries of men, and the breaking of bones"
(Exquemelin 1852, p. 472).

## 5. Analysis and Conclusions

Preparation indicates intent: eight of the twenty cases were planned and prepared
for in advance[23], more than mere boasting; for example, Low's men "always kept a Barrel
of Powder ready to blow up the Sloop rather than be taken" (*Tryals of Thirty-Six Persons
for Piracy* 1723, p. 9). These crews if caught promised that "they would set fire to the Ship
and blow themselves up; so that as they had lived, they would die together" (*A Select and
Impartial Account of the Lives, Behaviour, and Dying Words, of the Most Remarkable Convicts*
1740, vol. 2, p. 161), yet of those who made preparations, only Blackbeard's "resolute
Fellow" and Montauban's victims made the attempt. Teach's man was stopped in time, so
that despite their intentions, almost none made good on their threats. Others bragged of
their willingness to meet death before the opportunity arose to make good on it: Roberts'
men attempted to carry out their earlier threat—"they would blow up rather than be
taken"—but neither Roberts' pirates nor those of his lieutenant Skyrme were successful.

Boasts such as Spriggs' "if ever they should find themselves over-power'd, they would immediately blow their Ship up, rather than do Jolly Roger the disgrace to be struck" or Worley's "if he was overcome, would blow up the Conquerors", rang hollow in the end.

Of the eleven who attempted to blow up their ship, only four of the attempts were made directly on the Captain's orders: Blackbeard and Bonnet each sent a man to fire their powder but both were prevented, while the victims of the Salé Rovers and Montauban set off their traps successfully. All the other attempts were made out of desperation or vengeance—or fatalistic acceptance of their impending doom—on the initiative of just one or a few pirates. Whether their shipmates would have agreed varies: Bonnet's own men prevented their fellow pirate from killing them all, while Skyrme's crewman Roger Ball admitted that "John Morris fired a Pistol into the Powder, and if he had not done it, I would" (Johnson 1724, p. 269).

Given the lack of follow-through, were their threats merely boasts? We saw that pirates were no strangers to braggadocio. Roberts sent a letter to General Matthew of St. Kitts after Matthew burned Roberts' old *Royal Rover* and hanged some pirates: "for revenge you may assure yourselves here and hereafter not to expect anything from our hands but what belongs to a pirate", and warning Matthew not to harm another pirate then in his custody, "if we hear any otherwise you may expect not to have quarters to any of your Island" (Headlam 1933, p. 251). Yet of the fourteen pirates (out of twenty) who made the threat early and openly, wishing "for a blast to go to Hell with", only five made any sort of attempt[24], and none were successful.

And successes of any kind were indeed rare. The Salé Rovers' victims managed to destroy their attackers without sinking their own vessel; Montauban's English opponents succeeded in eliminating their attackers and the pirates' ship while keeping their own from being captured. The only pirate to blow himself up in the face of capture and an eventual "sour Look or two at choaking" was Joseph Cooper. Even newer pirate histories sometimes claim that "When the officers of the victorious vessel tried to board, Cooper and his mates blew up their ship, killing themselves" (Rediker 2004, p. 151); however, letters from *Diamond* confirm that Cooper succeeded only in killing himself, a few shipmates, and a handful of marines from the *Diamond*; his sloop the *Night Rambler* was even salvaged and converted for pirate hunting (*London Daily Post* 1726).

Threats of self-destruction among the Golden Age pirates were in part a way to intimidate prisoners, prey, and potential captors. They also reinforced the camaraderie among pirates who knew that in the absence of a general pardon, they may all face the noose if captured. The lack of follow-through on these threats and suicidal impulses, as well as the scant successes they produced among their rare uses, largely reduced the suicide threat from pirates to desperation, boast, and bluster.

**Funding:** This research received no external funding.

**Institutional Review Board Statement:** Not applicable.

**Informed Consent Statement:** Not applicable.

**Data Availability Statement:** Not applicable.

**Conflicts of Interest:** The author declares no conflict of interest.

## Notes

[1]    Depending on the author, the Golden Age might span as much as 1650 through 1730 (Powell 2018) or as little as 1716–1727 (Rediker 2004). For the purposes of this paper we will consider roughly the thiry-year stretch from the 1690s to the late 1720s with some allowances for related incidents on either side.

[2]    (Baer 2007, vol. 1, p. 321) Baer is quoting *The Daily Courant* of London, Friday 12 June 1724.

[3]    (Moseley 1790, p. 24) John Fillmore was the great-grandfather of the future U.S. President Millard Fillmore.

[4]    (Johnson 1724, p. 398) "Johnson" was a pseudonym; long thought to be Daniel Defoe, he may actually have been Defoe's publisher Nathaniel Mist. His *General History* is not always reliable but remains a useful period source where it can be corroborated

elsewhere. Warner's edition—the second of many editions of the *General History*—is an expanded version of a C. Rivington edition published earlier the same year.

5   (Johnson 1724, p. 230) This clause comes from the articles of Bartholomew Roberts.

6   (Fox 2013, p. 265) Fox notes further that authors such as Sherry and Pringle considered the provision of a pistol, powder, and shot for a marooned man to be an assumption of suicide but that there was also concurrent evidence that "marooned pirates were not sure to die on their islands".

7   (Burgess 2009, p. 150) For an example of juries displaying a lenience on pirates that authorities were loath to allow, captured pirates from the crew of the legendary Henry Avery were ordered to be tried a second time after an initial jury acquitted all of them.

8   (Blackstone 1775, vol. 4, p. 71) "*Hostis humani generis*" is usually rendered as "Enemies of all Mankind".

9   (Leeson 2009, p. 100) Leeson quotes from the *Boston News-Letter* of 16–23 June 1718.

10   (Johnson 1724, p. 244) The jack was a flag flown at the bow of a vessel as opposed to the ensign, which was a flag flown at the stern.

11   (Awtrey 1939) The "Pirates at Boston" were the survivors of the *Whydah* wreck which killed Samuel Bellamy.

12   (Rediker 2004, p. 149) Rediker is here quoting Hawkins' letter to the *The British Journal* dated 22 August 1724.

13   (Exquemelin 1852, p. 320) Translated editions of Exquemelin's work were often expanded and tended to favor the buccaneer heroes of the translator's country. The journal of Raveneau de Lussan was a later addition to Exquemelin's work, as was the Montauban chapter.

14   (*Tryals of Thirty-Six Persons for Piracy* 1723, p. 9) Powell was Harris' gunner but was on board Low's ship at the time; given the general suicidal threats made by Low, Lowther, Spriggs, Cooper, Harris, and other connected pirates, I have placed this under Low's name.

15   (*A Select and Impartial Account of the Lives, Behaviour, and Dying Words, of the Most Remarkable Convicts* 1740, vol. 2, p. 161) *Most Remarkable Criminals* is one of many similar compilations of pirate and highwayman biographies whose contents are often suspect. Letters between Gow and Fea are supposedly preserved and recorded; see Peterkin's *Notes on Orkney and Zetland* (Peterkin 1822).

16   (Johnson 1724, pp. 84–85) Johnson embellished and sensationalized the battle but Teach's self-destruction orders were confirmed in a 22 December 1718 letter from Virginia Lt. Governor Alexander Spotswood: "His [Teach's] orders were to blow up his own vessell if he should happen to be overcome, and a negro was ready to set fire to the powder, had he not been luckily prevented by a planter forced on board the night before and who lay in the hold of the sloop during the action of the pyrats". See Leigh's "Ghost of the Gallows" (op. cit.) for an exploration of "Caesar's" role.

17   (Grey 1933, p. 207) Grey is quoting Captain Carey, a victim of Roberts' whose troubles were mentioned by Gov. Samuel Shute of Massachusetts in a letter of 19 August 1720.

18   (*A Select and Impartial Account of the Lives, Behaviour, and Dying Words, of the Most Remarkable Convicts* 1740, vol. 2, p. 142) *Most Remarkable Convicts* again sensationalizes the incident but witness testimony from Gow's trial confirms the particulars.

19   (Shelvocke 1726, p. 12) Betagh was constantly at odds with Shelvocke and in his own book (also called *A Voyage Round the World*) downplayed all talk of mutinies and their seriousness.

20   (Johnson 1726, vol. 2, pp. 139–40) Johnson's second volume is less reliable than the first and contains some completely fictional accounts such as that of "Captain Misson" and the legendary pirate utopia of Libertalia. A number of modern pirate writers and historians mention this incident with 'Condent' but the source is inevitably Johnson; therefore, the details must be treated as suspect. Congdon/Condent himself was real and his career is otherwise fairly well documented; he survived to retire to Île de Bourbon.

21   (*Applebees Original Weekly Journal* 1720) Alternately *Mist's Weekly Journal* of the same date.

22   (Exquemelin 1852, p. 472) Montauban's account appears as an addendum to the 1699 edition of Exquemelin's work. It is heavily sensationalized and not well corroborated by other period sources.

23   De Graaf, Jean Bart, Montauban, Blackbeard, Worley, Low, Spriggs, and Gow.

24   Davis, Gow, Roberts, Congdon, and Harris.

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

Yesterday Arrived Here, Captain William Cross. 1726, *Daily Post*, June 23.