# Peer review of "“Merrily to Hell Together”: Threats of Self-Destruction among Golden Age Pirates"

_humanities, doi:10.3390/h11030071_

Round 1

Reviewer 1 Report

This paper is part of a growing and welcome trend in studies of pirate narratives; namely, a willingness to critique contemporary accounts of pirate activity and examine attitudes/prejudices that lie within them. The issue of suicide and self-harm among pirates has not, as far as I know, been sufficiently explored, and so this paper is a fine jumping-off point for further study of this topic. The "twenty cases" are all adequately described, as is the context of the pirates' legal status, especially in relation to matters of suicide.

That being said, I would urge the author to provide a bit more commentary on the twenty cases; their presentation at the moment tends to read somewhat like a laundry list of examples, especially near the end. It is important in pirate narratives to consider the sources of these anecdotes, and to question their trustworthiness. For example: in the case of the John Gow story, the anecdote about Williams threatening to blow up the ship derives from a somewhat dubious source, Edward Midwinter's History and Lives of all the Most Notorious Pirates (1725). The author(s) of the revised Gow chapter in the 1728 edition of Johnson's General History adopted the story from Midwinter, and slightly revised it. But the phrase "hellish Resolution" seems to have been added to the 1740 reprint from which the present author quotes--the addition of the adjective "hellish" is important for the overall effect on the reader. My point here is that some anecdotes, especially those derived from trial accounts, may be reliable, while others may be invented or embellished to serve the purpose of underlining the status of pirates as hostis humani generis. I understand that it is likely beyond the scope of the paper to analyse all twenty cases in such detail, but I would encourage the author to acknowledge the unreliability of these narratives more explicitly, if for no other reason than to encourage further study.

I might also encourage the author to consider Mark Hanna's article, “Well-Behaved Pirates Seldom Make History: A Reevaluation of the Golden Age of English Piracy.” (in Governing the Sea in the Early Modern Era: Essays in Honor of Robert C. Ritchie. Peter C. Mancall and Carole Shammas, eds.) Hanna provides a very useful counter-perspective to Rediker and Leeson, who tend sometimes to accept these narratives at face value. The author rightly notes that pirate threats of blowing up ships were often more bluster than anything else--but the proliferation of such stories in the narratives certainly colours readers' overall perception of pirates (something Hanna reconsiders). In the same way, the contrast of suicide as an additional punishment to marooning in the various Articles and the seemingly cavalier attitude towards suicide elsewhere might be worth underlining more clearly--it's mentioned, but I think more could be made of it.

One more small issue--if the author is quoting the edition of Johnson's General History published by Thomas Warner, it should be noted that Warner's is the second edition--which was substantially revised from the first edition. But I also applaud the author's using a particular edition rather than Schonhorn's very much outdated one.

Overall, I very much enjoyed this article, and I look forward to seeing its final version in (virtual) print.

Author Response

Many thanks for your comments and suggestions! I hope I have addressed them in the revised work.

 - edition of Johnson's General History published by Thomas Warner: clarified in the first Johnson footnote that this is the 2nd of many editions & was expanded from the 1st.

 - contrast of suicide as an additional punishment to marooning: addressed somewhat as part of Review 2's comments. The Articles do not necessarily assume suicide with the "one bottle of powder and shot" but various pirate historians claim this was the case. I expanded the footnote for Fox, who addresses this directly in "Piratical Schemes." 

 - provide a bit more commentary on the twenty cases: Agreed! Added some more context where possible. Also added at your suggestion the sources of some of these statements, either in the text if it was directly relevant or in the footnotes if if was more peripheral. Added mention of the colorfulness of these narratives to the paragraph on legal and societal views of suicide since I mentioned there how newspapers and the law portrayed them. 

 - notes on John Gow: Midwinter may have gotten his info from official records. Neil Rennie covers Gow in his 2013 book Treasure Neverland and mentions the Williams incident specifically, but as a source he cites the Examination of Peter Rollson, 2 April 1725, HCA, 1/55, fo. 108r, PRO. I don't have access to that document but we know Johnson and others borrowed, revised, paraphrased, and stole from trial docs, public records, interviews, etc., so it's possible Midwinter did the same for Gow's story (as you point out, ironically later copied by 'Johnson'). Maybe someone with access to PRO folders could verify if Rollson's deposition really covers the Gow/Williams incident? Rennie's Gow chapter also cites the examinations of William Melvin ( 3 April 1725, HCA, 1/55, fo. 116v, PRO), James Belbin, (2 April 1725, HCA,1/55, fo.107v, PRO) and John Gow / Smith himself ( 2 April 1725, HCA, 1/55, fo. 105v, PRO), among others, so there's definitely more info out there somewhere. Gow's letters to Fea with the "as they had lived, they would die together" quote are more suspect. I added footnotes to the Gow entries reflecting both of these.

Reviewer 2 Report

It would be helpful to understand a little more about contemporary reactions to suicide (especially in the light of Spriggs and his companion planning to shoot each other, rather than each shoot himself – thus technically avoiding self-destruction).

The article makes its point. The list below deals with stylistic issues / points that would benefit from being expressed more clearly.

30. Should ‘may’ be ‘might’

31. Should ‘Pirates’ have an initial capital?

36. It would be helpful to give the dates of ‘the Golden Age period’, especially as (197-8) one example is said to be earlier

42. Should the quotation be referenced?

46. After stating ‘pirates were intimately familiar with suicide’ the reader expects an example. The following Snelgrove anecdote does not give an example of pirate suicide. The Spriggs example is closer  – this  could be tightened up.

63. The assumption at this point is that John Phillips committed suicide but it turns out that it was ‘Phillips’ man’ (71).

77. Is the shot really so that they could end their lives? At best this is ambiguous – they may have used the shot to hunt for food. In fact this is eludidated (83-5) so it’s annoying to have it so plainly stated at first, only to have it qualified later.

85-8. The point really isn’t clear.

Fn 11. This reference to the trial of Avery’s crew just muddies the waters because the jury was not passing judgement on a suicide.

Fn 14. It would be good to give specific page references, not just the page extent of the article (especially as the article also deals with pirates as fiends not just pirates as cowards and fools)

210. the term ‘corsair’ is usually reserved for the Barbary corsairs of North Africa

223. ‘though his cruelty multiplied after he learned that Harris had been executed’) This seems a non sequitur.

252. This is not a good way to start a paragraph as the reader does not find out who the subject is until 4 lines later.

341. Naval warships at this period were not called HMS (& 418)

342. ‘Diamond (singular) reported … they’ (& 344) is needlessly inelegant

Author Response

Many thanks for your review! I hope I have addressed your concerns below.

30. changes to "might"

31. Pirates changed to pirates

36. footnote added re: Golden Age dates

42. This short quote is mainly for flavor; it's a snippet of a longer quote referenced in full later in the paper so I have not footnoted the short version here.

46. I switched this paragraph and the following one to make it clearer where pirates embraced suicide vs. embracing death by other violent means. 

63. Edited the sentence to clarify that the suicide was one of Phillips' pirates, not Phillips himself.

77. I reworded this sentence and expanded the Fox footnote; Fox addressed this directly, that marooned men were not automatically expected to commit suicide, but that some historians did assume this to be the case.

85-8. This was meant to lead into the next section convering legal and societal attitudes toward suicide in general, which itself leads into legal views of pirates as beyond legal protection. Reworded slightly to tie in better with the next paragraph.

Fn 11. Clarified the footnote: it was supposed to be an example of juries' willingness to acquit pirates (via "jury nullification") just as they were willing to excuse suicides.

Fn 14. Typo. Page range fixed. 

210. Corrected "corsair" (Barbary raiders) to "corsaire" (equates in French use to a privateer).

223. Comment about Low noted. I kept it here to add a little flavor to Low and his hypocrisy. Reworded to make his relucatance toward self-destruction the focus instead.

252. Reordered to make clear from the start that this is about Roberts.

341. Naval warships at this period were not called HMS (& 418) - Removed the two references to HMS Diamond but left the other HMS references. The same error is repeated in much of the most common pirate history literature (Woodard, Marley, Rediker, etc.), even about Diamond

342. Reworded for less clunkiness.